

# Assessing reliability and accuracy of qPCR, dPCR and ddPCR for estimating mitochondrial DNA copy number in songbird blood and sperm cells

Laima Bagdonaitė[1], Erica H. Leder[1,2,3], Jan T. Lifjeld[1], Arild Johnsen[1] and Quentin Mauvisseau[1]

[1] Natural History Museum, University of Oslo, Oslo, Norway
[2] Tjärnö Marine Laboratory, Department of Marine Sciences, University of Gothenburg, Strömstad, Sweden
[3] Section of Ecology and Evolution, Department of Biology, University of Turku, Turku, Finland

## ABSTRACT

Mitochondrial DNA (mtDNA) copy number varies across species, individuals, and cell types. In birds, there are two types of cells with a relatively low number of mitochondria: red blood cells and spermatozoa. Previous studies investigating variation of mitochondrial abundance in animal sperm have generally used quantitative PCR (qPCR), but this method shows potential limitations when quantifying targets at low abundance. To mitigate such issues, we investigated and compared the reliability and accuracy of qPCR, digital PCR (dPCR) and droplet digital PCR (ddPCR) to quantify high and low concentration DNA. We used synthetic DNA targets, to calculate the limit of detection and the limit of quantification and found that with both dPCR and ddPCR, these limits were lower than with qPCR. Then, to compare quantification accuracy and repeatability, we used DNA extracted from blood and sperm cells of Eurasian siskin. We found that qPCR, dPCR and ddPCR all reliably quantified mitochondrial DNA in sperm samples but showed significant differences when analyzing the typically lower levels of mtDNA in blood, with ddPCR consistently showing lower variation among replicates. Our study provides critical insights and recommendations for future studies aiming to quantify target mtDNA and indicates that dPCR and ddPCR are the preferred methods when working with samples with low abundance of mtDNA.

# INTRODUCTION

Mitochondria are double-membraned organelles commonly referred to as the "powerhouse of the cell" and are found in most eukaryotic cell types (*McBride, Neuspiel & Wasiak, 2006*). The interactions between mitochondrial and nuclear genomes enable oxidative phosphorylation (OXPHOS) processes *via* which adenosine triphosphate (ATP) is produced, providing energy supporting various cell processes (*Cavalier-Smith, 1987*; *Stier et al., 2013*; *Hill, 2017*). Due to variation in metabolic needs, mitochondria numbers vary tremendously between species, individuals and different cell types

Corresponding author
Laima Bagdonaitė,
laima.bagdonaite@nhm.uio.no

(*Cole, 2016*; *Reverter et al., 2016*; *Soledad, Charles & Samarjit, 2019*). For example, the brain and different muscle tissues (often associated with higher metabolic needs), systematically show higher mtDNA copy number than other tissues or cells, such as red blood cells, which typically lack mitochondria in most mammals and are present only in small amounts in birds (*Stier et al., 2013*; *Reverter et al., 2016*; *Li et al., 2023*). Likewise, sperm cells are known to have a much lower mitochondrial content than somatic cells (*Song & Lewis, 2008*).

In the animal kingdom, sperm cells consist of three key components: the head, midpiece and tail, with the midpiece harboring the mitochondria (*Jamieson, 2007*). Notably, bird spermatozoa have been found to contain variable amounts of mitochondria per cell, with numbers ranging from 20 to 2,500 mitochondria, and display large inter- and intraspecific variation (*Jamieson, 2007*). Passerine birds especially are known for their tremendous variation in sperm morphology, including differences in the shape and volume of different sperm components (*Pitnick, Hosken & Birkhead, 2009*). While previous studies have focused on the relationship between sperm midpiece morphology and various sperm properties, such as ATP production, swimming speed or sperm competition (*e.g.*, *Lüpold et al., 2009*, *2020*; *Rowe et al., 2013*; *Cramer et al., 2021*), there is a scarcity of studies focused on quantifying mitochondrial DNA content in passerine birds such as in (*Knief et al., 2021*). Most previous studies investigating mitochondrial abundance in animal sperm, such as in common fruit fly (*Drosophila melanogaster*) (*DeLuca & O'Farrell, 2012*), Chinook salmon (*Oncorhynchus tshawytscha*) (*Wolff & Gemmell, 2008*), bull (*Bos taurus*) (*Nguyen et al., 2023*) and zebra finch (*Taeniopygia guttata*) (*Knief et al., 2021*), used quantitative polymerase chain reaction (qPCR) techniques.

Quantitative PCR monitors the amplification of the targeted molecule in real time through fluorescence scanning at the end of each amplification cycle, and through the additional analysis of a calibration curve of samples with a known concentration, quantifies the amplicon of interest (*Pfaffl, 2019*). A wide range of studies have used qPCR to answer various research questions due to its speed, relative sensitivity and availability (*Kuang et al., 2018*; *Bommerlund et al., 2023*; *Cawthon, 2002*). However, qPCR shows limitations due to the necessity of generating standard curves to quantify samples and has been associated with larger variability when analyzing samples at low concentration (*Mauvisseau et al., 2019a*).

Recently, there has been an increase in the utilization of absolute quantification methods, such as digital PCR (dPCR) and droplet digital PCR (ddPCR). Both methods rely on the partitioning of an initial reaction mixture into thousands of independent droplets, each becoming a separate PCR reaction, in such a way that partitions contain one or a few target molecules or no target at all (*Pinheiro et al., 2012*; *Hindson et al., 2013*; *AMC Technical Brief, 2017*; *Quan, Sauzade & Brouzes, 2018*). One major difference between the partitioning of the sample in the two methods is the use of microfluidic techniques such as micro well plates in dPCR, and the use of oil and emulsification in ddPCR (*Hindson et al., 2013*; *Quan, Sauzade & Brouzes, 2018*). With both dPCR and ddPCR, following end point PCR, fluorescence signals of each partition are measured to ensure the presence or absence of the targeted amplified DNA molecule. Then, using Poisson statistics, the ratio of positive

to the total number of partitions is used to calculate the absolute target concentration (copies/µL) (*AMC Technical Brief, 2017*; *Quan, Sauzade & Brouzes, 2018*). The last few years have seen commercialization of dPCR and ddPCR technologies with an increase in scientific studies using these methods for quantification of target nucleic acids (*Elmahalawy et al., 2018*; *Mauvisseau et al., 2019b*), as well as screening for pathogen DNA and RNA in clinical samples with low amount of nucleic acids, like serum or swabs (*Stenzel et al., 2024*; *Stenzel et al., 2019*; *Łukaszuk, Dziewulska & Stenzel, 2024*). The dPCR method has also been applied to mtDNA in human spermatozoa, revealing that mtDNA number has strong implications for sperm quality and fertilization success (*Boguenet et al., 2022*). With the increasing research focus, it seems likely that the quantification of low concentration targets, including mtDNA will continue to be of high importance in the future, both in natural sciences and in human health studies. While digital PCR methods have been argued to be more reproducible than qPCR before (*Huggett et al., 2013*), to our knowledge, no study has investigated the trade-offs between qPCR, dPCR and ddPCR to quantify target mtDNA molecules at low and high concentration.

In this study we adopt a comparative approach, examining one conventional qPCR method and two absolute quantification methods, dPCR and ddPCR. We aim to evaluate the sensitivity and accuracy of quantifying target mtDNA (Fig. 1) across methods and to identify their quality of results (see four potential scenarios in Fig. 1). We first used a synthetic mtDNA fragment to determine the limit of detection (LOD) and limit of quantification (LOQ) of each method. We then compared the mtDNA quantification results and associated variation across replicates in two distinct types of mitochondria-carrying avian biological samples: blood and sperm, associated respectively with low and moderately higher mtDNA numbers. An earlier version of the manuscript is available as preprint on bioRxiv: https://doi.org/10.1101/2024.11.08.622696.

## MATERIALS AND METHODS

### Primers and synthetic DNA

We designed a new set of primers targeting a range of passerine species and a fragment of their mitochondrial 16S gene. The primer sequences (Forward 5′-ATTATTGAGCGAA CCCGTCTC-3′ and Reverse 5′- TTCACAGGCAACCAGCTATC-3′) were designed following the method described by *Amer, Ahmed & Shobrak (2013)*. In brief, we downloaded multiple mitochondrial 16S sequences from a range of songbird species (infraorder Passerides), covering several genera (*Spinus spinus*, *Emberiza schoeniclus*, *Coccothraustes coccothraustes*, *Fringilla coelebs*, *Pyrrhula pyrrhula*, *Parus major*, *Luscinia svecica*) from GenBank (http://www.ncbi.nlm.nih.gov) (Table S1). These sequences were then aligned in the Geneious Pro R10 software (https://www.geneious.com) with the Clustal Omega multiple alignment function and consensus sequences for each species were generated. Then, the consensus sequences were aligned, and primers were created using the primer design function. The specificity of the newly designed primers was visually assessed by inspecting the multi-species alignment and confirmed using the Integrated DNA Technologies (IDT) PrimerQuest tool (PrimerQuest™ program; IDT, Coralville, Iowa, USA). Additional validation was conducted using the primer blast function from the

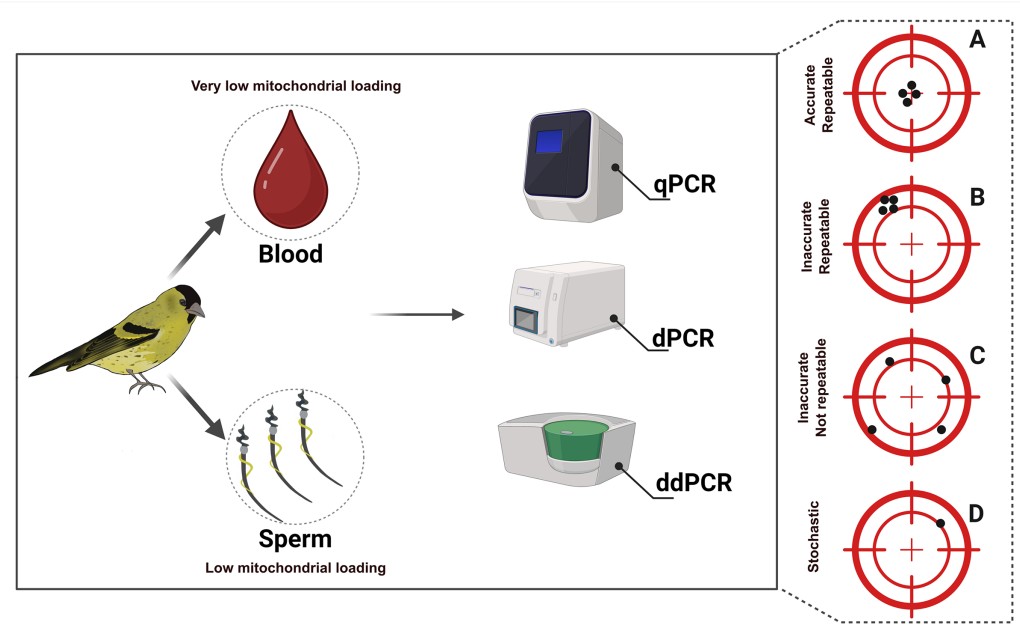

**Figure 1 Graphical illustration of the study design.** Eurasian siskin (*Spinus spinus*) was selected as the model organism for the study. DNA was extracted from 10 blood and 10 sperm samples of different male individuals. Each sample was quantified in triplicate across all three PCR platforms. The panel on the right (A–D) indicates what level of reliability can be expected when running such assays. (A) An optimal scenario with accurate and repeatable target quantification across replicates; (B) target quantification is not accurate, but is repeatable across replicates; (C) target quantification is not accurate, nor repeatable across replicates; (D) target DNA is amplified in only one replicate-target detection is stochastic across replicates and quantification is incorrect, leading to unreliable results. Figure created using BioRender.          

NCBI website (https://www.ncbi.nlm.nih.gov/tools/primer-blast/). Following these *in silico* validation steps, we performed an additional *in vitro* validation with PCR and ddPCR on DNA extracted from several songbird species: *S. spinus, Turdus pilaris, Phylloscopus trochilus, L. svecica, Fringilla montifringilla, P. pyrrhula, Regulus regulus, C. coccothraustes, E. schoeniclus*. We used DNA previously extracted from sperm samples and stored at the DNA bank at the Natural History Museum of Oslo (NHMO). Sample information is provided in Table S2. These thorough validation steps, including temperature gradient analysis, were performed to ensure the specificity and optimal amplification of the target fragment.

Additionally, we designed a synthetic DNA fragment of mitochondrial 16S (5′-ATTA TTGAGCGAACCCGTCTCTGTGGCAAAAGAGTGGGATGACTTGTTAGTAGTGG TGAAAAGCCAATCGAGCTGGGTGATAGCTGGTTGCCTGTGAA-3′), following a method described by *Han et al. (2023)*. We used the same multiple sequence alignment as described above, and the consensus sequence was created with most frequent nucleotide for each base, containing no ambiguous nucleotides—only A, T, C, G. Then, we tested this sequence against our previously designed primers by using "Test with Saved Primers" function in Geneious Pro R10 software, which ensured the match between primers and the fragment of our gene of interest. Primer forward and reverse sequences, as well as 16S fragment were ordered from Integrated DNA Technologies (IDT, Inc., Coralville, Iowa,

USA). We used this synthetic fragment of mtDNA to determine limits of detection and quantification with the three methods used in this study.

## Samples

We selected the Eurasian siskin (*Spinus spinus*) as our study species because they are common breeding birds in Norway and are easily caught at bird feeders. Siskins have relatively long sperm cells (219 μm), carrying long midpieces (190 μm) (*Omotoriogun et al., 2020*), which could indicate they are assembled from a high number of mitochondria (*Knief et al., 2021*). Blood and sperm samples used in this study were collected from male birds during the breeding seasons 2021–2022. A total of 10 blood samples and 10 sperm samples were collected from five individuals in 2021 and 13 in 2022 (Table S3). Birds were caught using mist nets at their breeding sites in southern Norway and released at the trapping site immediately after sampling. Sperm samples were collected *via* cloacal massage in a similar way as described in *Wolfson (1952)*. Two males were sampled twice. The ejaculate was collected in a micro-capillary tube and immediately mixed with phosphate-buffered saline (PBS) (*Kleven et al., 2008*). This was done to prevent the sperm cells from clumping together and to enable pipetting. Sperm samples were then purified through a series of centrifugation and washing with PBS, and frozen at −80 °C until DNA extraction. Blood samples were collected through brachial venipuncture (10–20 μL) and stored in 96% ethanol. Sampling was conducted in adherence to ethical guidelines for use of animals in research and with permission from all relevant local authorities and approved by the Norwegian Food Safety Authority (permit no. 23294 and 29575) and The Norwegian Environment Agency (permit no. 2021/39021).

DNA was extracted from blood samples using the E. Z. N. A. Tissue DNA kit (Omega Bio-Tek, Inc, Norcross, GA, USA) following the manufacturer's guidelines. DNA was extracted from sperm samples using the QIAamp DNA Micro Kit (Qiagen, Inc. Valencia, CA, USA) following a protocol for sperm samples as described by *Kucera & Heidinger (2018)*. We made two changes from the protocol; (i) the mixture of the sample and dithiothreitol (DTT) buffer was incubated for 2 h in order to ensure the proper lysis of sperm cells and the release of DNA and (ii) the final elution was carried out with 25 μL of MilliQ water and repeated twice to ensure a maximum DNA yield. Following DNA extraction, total DNA concentration from each sample was measured with Qubit 2.0 fluorometer with the DNA High Sensitivity Assay (Invitrogen, Waltham, MA, USA). Sample concentrations were then normalized to a concentration of 0.5 ng/μL using MilliQ water before further downstream analysis. All DNA extractions were performed in a pre-PCR dedicated laboratory, all the bench surfaces were wiped with 5% deconex solution and absolute ethanol before and after the procedures.

## qPCR

Samples were analyzed on a Bio-Rad CFX96 Real-Time System (Bio-Rad Laboratories, Hercules, CA, USA) using the primers described earlier. Quantitative PCR reactions were conducted in a 20 μL final volume containing 10 μL QIAcuity EvaGreen Mastermix (Qiagen, Hilden, Germany), 8.5 μL MilliQ water, 0.25 μL of each primer (10 μM) and 1 μL

of template DNA at 0.5 ng/µL (see the sample description above). The qPCR program was as follows: initial denaturation at 95 °C for 2 min 10 s, followed by 40 cycles of denaturation at 94 °C for 15 s, annealing at 55 °C for 15 s and extension at 72 °C for 15 s. This was followed by 1 min at 72 °C and held at 4 °C until the amplified samples were removed from the qPCR machine. The plate with sperm and blood samples also included the standard dilutions which were used to calculate starting concentrations of the DNA samples. The change in fluorescence intensity was measured at the end of each extension step, and results were analyzed using the CFX manager software (version 3.1; Bio-Rad, Hercules, CA, USA). Quantification cycle (Cq) values were determined by manually establishing a threshold line within the exponential amplification phase across both amplification plots.

## dPCR

Samples were analyzed on a nanoplate-based QIAcuity Digital PCR System (Qiagen, Hilden, Germany) using the automated workflow provided by the QIAcuity Software Suite (v.2.1). dPCR reactions were performed in a 10 µL final volume consisting of 3.33 µL QIAcuity EvaGreen Mastermix (Qiagen, Hilden, Germany), 5.44 µL MilliQ water, 0.13 µL of forward and reverse primers (10 µM) and 1 µL template DNA at 0.5 ng/µL (see the sample description above). dPCR reactions were then dispensed to 96-well 8.5k QIAcuity Nanoplates (Qiagen, Hilden, Germany), to divide all single samples into 8,500 of individual partitions before end-point PCR. The dPCR plates used in this study resulted in half as many partitions as ddPCR. Reaction conditions were as follows: 2 min initial denaturation at 95 °C, followed by 35 cycles of denaturation for 15 s at 95 °C, 15 s annealing at 55 °C, and 15 s extension at 72 °C, with a final step of 5 min at 40 °C. Results were analyzed using the QIAcuity Software Suite (v.2.1.8.23; Qiagen, Hilden, Germany). Green channel was used to detect the EvaGreen® fluorophore. Exposure duration was 500 ms for sperm and blood samples, and 300 ms for the synthetic mtDNA fragment. The common fluorescence intensity (RFU) threshold to identify positive and negative samples was set at 57 for the analysis of sperm and blood samples, and at 30 for synthetic mtDNA analysis.

## ddPCR

Samples were analyzed on a Bio-Rad QX200 ddPCR System. ddPCR reactions were performed in a 20 µL final volume, consisting of 10 µL Bio-Rad ddPCR EvaGreen supermix, 0.25 µL of forward and reverse primers (10 µM), 8.5 µL of MilliQ water and 1 µL template DNA at 0.5 ng/µL (see the sample description above). After vigorous mixing, each reaction was pipetted into the sample well of a DG8 Droplet Generator Cartridge, and 70 µL of Droplet Generation Oil for EvaGreen was added to the oil wells. Droplets were then generated using the QX200 Droplet Generator (Bio-Rad, Hercules, CA, USA), and a final 40 µL volume of droplets for each sample was carefully transferred to a ddPCR 96-well plate, later sealed with pierceable foil using a PX1 PCR Plate Sealer (Bio-Rad, Hercules, CA, USA) before end-point PCR. The specifications of the system ensure the partitioning of the sample into approximately 20,000 droplets, although the final number

of partitions can vary between the runs and can result in fewer than 20,000 droplets (see results). PCRs were performed on a BioRad CFX96 Real-Time System (Bio-Rad Laboratories, Hercules, CA, USA). ddPCR conditions were as follows: 10 min at 95 °C, followed by 40 cycles of denaturation for 30 s at 94 °C and annealing at 55 °C for 1 min, with ramp rate of 2 °C/s, followed by 10 min at 98 °C and a hold at 8 °C. Droplets were then read on a QX200 droplet reader (Bio-Rad, Hercules, CA, USA). Quantification data were checked using the Bio-Rad QuantaSoft software (v.1.7.4.0917). Thresholds for positive signals were determined according to QuantaSoft software instructions, and all droplets above the fluorescence threshold (10,000) were counted as positive events, those below it being counted as negative events.

## Estimation of the limit of detection and limit of quantification

The MIQE guidelines (*Bustin et al., 2009*) define the LOD as the lowest concentration at which 95% of the positive samples are detected. The definition of the LOQ varies between different studies, but is usually defined as the lowest concentration at which replicates show a CV of or less than 35% (*Forootan et al., 2017*; *Brys et al., 2021*). To establish the LOD and the LOQ, we performed a 10-fold dilution series with the synthetic mtDNA fragment (see the primers and synthetic DNA section). The synthetic DNA was used to avoid nonspecific amplification and to ensure sensitivity of the PCR assays (*Han et al., 2023*). The dilution series ranged from 2.50E−04 ng/µL to 2.50E−12 ng/µL, and included nine dilution points.

Ten technical replicates of each dilution point were analyzed with each quantification platform, and a minimum of four negative controls were included on each plate. The serial dilution further allowed us to quantify the mtDNA detected with qPCR using the standard curve generated. The LOD for ddPCR, dPCR and qPCR were calculated using dedicated R scripts as in *Hunter et al. (2017)*. We also followed the method outlined by *Klymus et al. (2020)* to assess the LOD and LOQ for qPCR assays. Thus, the LOD for qPCR was calculated using both methods (*Hunter et al., 2017*; *Klymus et al., 2020*) to assess whether they will return similar results. Both scripts resulted in the same LOD for qPCR. The LOQ for ddPCR and dPCR were estimated following the threshold and methods described in *Forootan et al. (2017)*, *Brys et al. (2021)*.

## Statistical analyses

In the qPCR analysis, the concentration of copy number per µL was estimated using the standard curve generated from the serial dilution of the synthetic mtDNA fragment (*Bustin et al., 2009*). In the dPCR and ddPCR assays, the copy number per µL was calculated using the QIAcuity Software Suite and QuantaSoft respectively, using the fraction of positive and negative partitions based on Poisson distribution (*Coudray-Meunier et al., 2015*). Coefficient of Variation (later referred to as CV) was calculated for quantification results as the ratio between the standard deviation and the mean multiplied by 100. Smaller CV values across replicates indicate better repeatability. All statistical analyses were performed in R v4.3.2 using RStudio/2023.12.0+369 and the packages tidyverse v2.0.0 (*Wickham et al., 2019*), and vegan v2.6-4 (*Oksanen et al., 2022*). The

methods were compared using several statistical tests: we compared the results of the synthetic mtDNA quantification with dPCR and ddPCR using Welch t-test to allow for unequal variance. We compared the sperm and blood mtDNA quantification results using one-way analysis of variance (ANOVA), followed by Tukey's Honest Significant Difference test if significant. Plots were generated using ggplot2 v3.4.4 (*Wickham et al., 2016*).

## RESULTS

The primers developed in this study were found to amplify mtDNA across a range of passerine species and performed well across qPCR, dPCR and ddPCR. The negative controls did not amplify with either method, except for one replicate during the analysis of the serial dilution using synthetic mtDNA to establish the LOD and LOQ on the ddPCR platform, likely due to a pipetting error. The average number of partitions generated was 8,050 with dPCR and 16,082 with ddPCR (see Table S6 for all quantification information). These numbers follow the manufacturers' specifications (see details provided in Methods section). Both dPCR and ddPCR platforms could not reliably quantify high concentration of synthetic DNA, as both platforms reached saturation due to excessive numbers of positive partitions. For this reason, we excluded high concentrations of the dilution series (*i.e.*, those dilutions which led to the number of positive partitions exceeding 95%) from the downstream analysis. In contrast, qPCR was not able to detect and quantify the three lowest dilution points (Cq > 39), therefore these lower dilutions were omitted from the analysis.

### LOD and LOQ

qPCR Efficiency % ranged from 94.5 to 116.3, had slope from −3.462 to −2.984, had Y intercepts from 7.411 to 38.623, and had $R^2$ from 0.968 to 0.976 for synthetic and biologically-derived mtDNA, respectively. For qPCR, the LOD was established at 2.42 copies/µL and LOQ was established at 4.9 copies/µL. dPCR had an LOD at 0.51 copies/µL and LOQ at 2.12 copies/µL, while ddPCR had an LOD at 0.63 copies/µL and LOQ at 1.23 copies/µL (Table S4 and Figs. S1–S6).

### Synthetic mtDNA

We investigated the correlation between the measured concentration and the expected concentration of the dilution series (Fig. 2; Table S5). Only dPCR and ddPCR were compared in this step, as the dilution series was used to convert qPCR Cq values into concentrations. Based on the results of the Welch t-test, mtDNA quantification did not significantly differ between dPCR and ddPCR at five of seven dilution points (Table 1). We found that ddPCR had a lower CV than dPCR at five of seven concentrations of synthetic mtDNA.

### Sperm and blood mtDNA

The comparison of copy numbers estimated using the three methods (Fig. 3) revealed similar mean sperm mtDNA copy number with dPCR and ddPCR: 310.81 copies/µL and 298.26 copies/µL, respectively, while the copy number measured with qPCR was

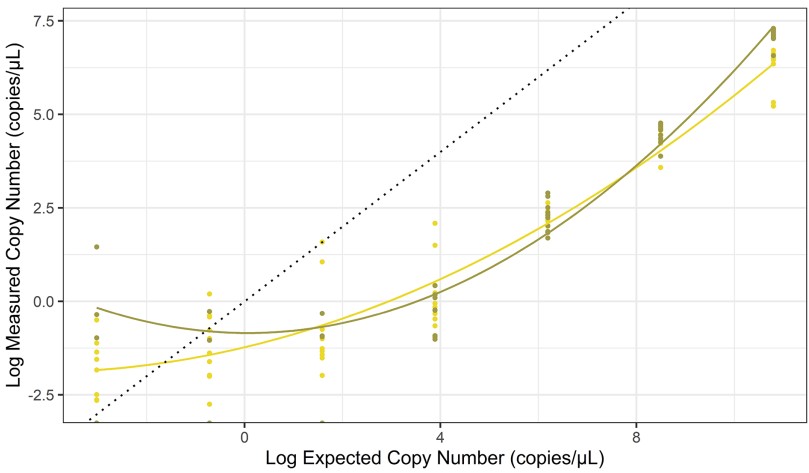

**Figure 2 The relationship between log-transformed expected copy number of the synthetic DNA and log-transformed copy numbers obtained with dPCR (olive green) and ddPCR (yellow).** The figure shows data from seven dilution points (concentration range 2.50E−06 to 2.50E−12 (ng/µL)). The black dotted line shows where the expected concentration is equal to the observed concentration. Polynomial regression lines were fitted to the concentrations.

**Table 1 Comparison of synthetic mtDNA quantification, variability (CV%), and statistical significance (T-test p-values) across dilution steps using dPCR and ddPCR.**

| Dilution step | Copy number, copies/µL | | Coefficient of variation (%) | | T-test |
|---|---|---|---|---|---|
| ng/µL | dPCR | ddPCR | dPCR | ddPCR | $p$ |
| 2.50E−06 | 1,247.20 | 543.10 | 18 | 46 | 3.44E−06 |
| 2.50E−07 | 89.66 | 82.51 | 24 | 22 | 0.35 |
| 2.50E−08 | 10.30 | 9.73 | 42 | 19 | 0.71 |
| 2.50E−09 | 0.81 | 2.02 | 52 | 119 | 0.15 |
| 2.50E−10 | 0.15 | 0.99 | 172 | 162 | 0.14 |
| 2.50E−11 | 0.11 | 0.42 | 228 | 85 | 0.04 |
| 2.50E−12 | 0.54 | 0.19 | 250 | 96 | 0.43 |

**Note:**
Welch t-test was used to compare the concentrations obtained with both methods.

higher−507.13 copies/µL. The comparison of measured blood mtDNA concentration across the three platforms, showed more differences; mean concentrations measured with qPCR, dPCR and ddPCR were as follows: 17.19 copies/µL, 21.84 copies/µL and 36.14 copies/µL. There was no significant difference in sperm mtDNA quantification across the three methods (ANOVA, $p$ = 0.25). However, blood mtDNA quantification was significantly different (ANOVA, $p$ = 3.33E−06). The one-way ANOVA was followed by Tukey's HSD test, which revealed no statistically significant difference between qPCR and dPCR (adjusted $p$ = 0.3). In contrast, ddPCR differed significantly from both dPCR and qPCR (adjusted $p$ = 2.1E−04 and 3.7E−06, respectively).

We calculated CV for each individual using the mtDNA copy number measured for each technical replicate across qPCR, dPCR and ddPCR (Fig. 4). Mean CV across methods was <50%. In sperm samples, mean CV was 23% for qPCR, 25% for dPCR, and 9% for

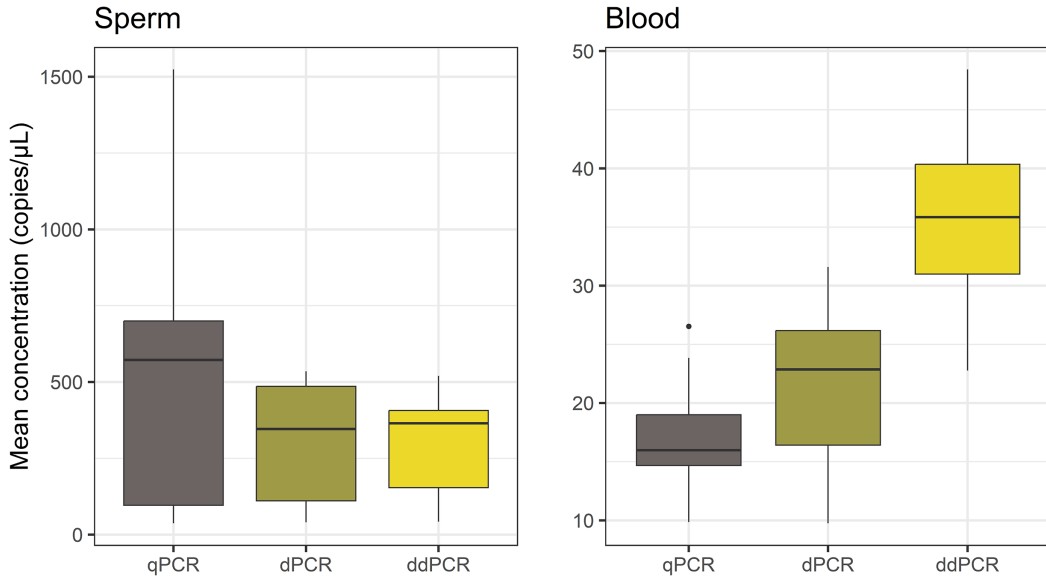

**Figure 3 Comparison of qPCR, dPCR and ddPCR quantification of mtDNA copy number in sperm and blood samples.** Each box plot represents mean concentrations obtained for 10 biological samples for the two types of cells. Sperm mtDNA quantification was not significantly different with the three methods, while blood quantification with ddPCR was significantly different from the other two methods, and there were no significant differences between qPCR and dPCR.

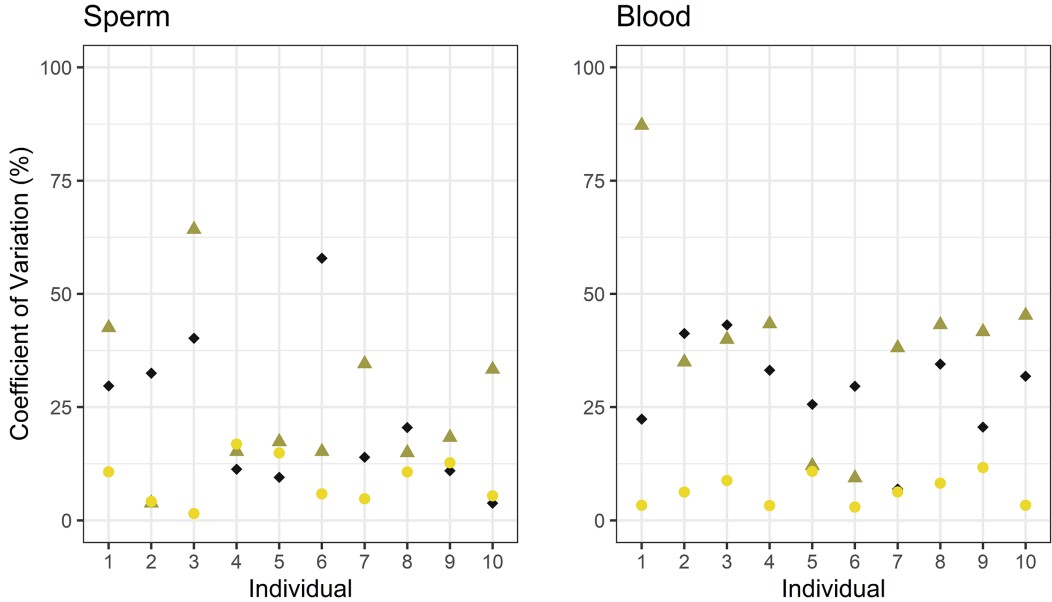

**Figure 4 Comparison of coefficient of variation in mtDNA copy number estimation in sperm and blood across samples, every sample was quantified in triplicate.** qPCR (black diamond), dPCR (olive green triangle) and ddPCR (yellow circle).

ddPCR, while blood samples had a higher CV with qPCR (30%) and dPCR (40%), and lower with ddPCR (7.5%). The CV differed significantly across the three methods in both sperm and blood samples (ANOVA, $p = 0.03$ and 4.31E−05, respectively). Tukey's HSD test, applied to the one-way ANOVA results, indicated no significant difference in CVs

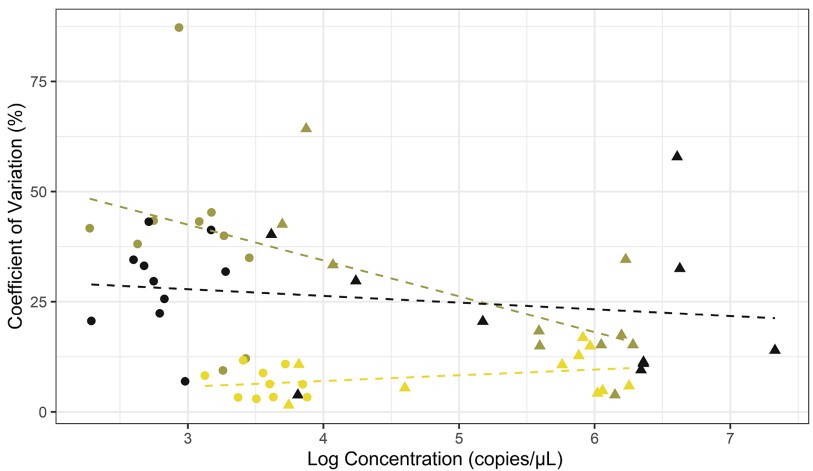

**Figure 5** **The relationship between coefficient of variation (CV) and log-transformed mean mtDNA concentration measurement of blood and sperm samples across the three quantification platforms.** Regression lines for each method were added to show how CV changes with the increase of concentration ● blood samples; ▲ sperm samples. Black—qPCR, olive green—dPCR, yellow—ddPCR. Overall, the lowest CV is exhibited by ddPCR.

between qPCR and dPCR for sperm (adjusted $p$ = 0.89) or blood samples (adjusted $p$ = 0.22). In contrast, ddPCR had lower CV than qPCR and dPCR in sperm samples (adjusted $p$ = 0.088 and 0.034, respectively) and in blood samples (adjusted $p$ = 3.24E−03 and 3.45E−05, respectively).

Finally, we explored the relationship between the CV in blood and sperm samples and the mean measured mtDNA concentration across the three methods (Fig. 5) and found that CV decreased as the concentration increased. Lower CVs were obtained when quantifying mtDNA in sperm samples, especially for qPCR and dPCR.

## DISCUSSION

In this study, we investigated the accuracy and repeatability of qPCR, dPCR and ddPCR quantification platforms to assess their reliability in quantifying mtDNA targets in avian blood and sperm samples. We first investigated the LOD and LOQ of each method using a 10-fold serial dilution of synthetic mtDNA (*Pfaffl, 2019*), and while our LOD and LOQ measurements for qPCR, dPCR and ddPCR are in agreement with other previously published studies (*Forootan et al., 2017*; *Deprez et al., 2016*), using a five-fold dilution series with more dilution points could increase the measurements at lower concentration, therefore further improving the resolution of LOD and LOQ. The highest concentration of the dilution series led to saturation of both dPCR and ddPCR platforms, therefore impeding their measurements. This was not the case for qPCR, suggesting that qPCR might be more reliable when quantifying mtDNA in highly concentrated samples. However, we could not detect the target mtDNA of the three lowest concentrations from the dilution series (4.8, 0.48 and 0.048 copies/μL) using qPCR. This was not the case for dPCR and ddPCR, which seem more useful for quantifying low-concentration samples.

**Table 2 Comparison of the three quantification methods and recommendation for usage.**

| Method | Requires calibration curve | Input DNA concentration | | |
|---|---|---|---|---|
| | | Low | Medium | High |
| qPCR | Yes; relative quantification | × | ✓ | ✓ |
| dPCR | No; absolute quantification | ✓ | ✓ | × |
| ddPCR | No; absolute quantification | ✓* | ✓* | × |

**Note:**
  * ddPCR has higher repeatability (lower CV).

This is in accordance with various studies highlighting the higher sensitivity of both dPCR and ddPCR compared to qPCR (*Mauvisseau et al., 2019b*; *Tellinghuisen, 2020*; *Park et al., 2021*).

    We found a higher copy number in sperm mtDNA compared to blood mtDNA. This can be attributed to a greater amount of mtDNA present in sperm cells compared to bird blood cells, in which mtDNA is detectable only at very low levels (*Reverter et al., 2016*). This accounts for the nearly tenfold difference in mtDNA concentrations we observed between the two types of cells. When comparing the three methods to measure concentrations of sperm samples that are known to have a relatively higher mtDNA copy number, we found no significant differences between platforms. mtDNA was quantified efficiently in all sperm samples even when a low amount of input DNA (0.5 ng) was used in the reactions, which is important when working with wild birds with variable and often rather low volumes of ejaculates (reviewed in *Gee et al., 2004*). However, we noticed lower variation between replicates with ddPCR than with the other two methods. In contrast, blood samples have a lower mtDNA content, and even though all three methods were able to quantify the target mtDNA, ddPCR exhibited a much lower CV compared to qPCR and dPCR and thus seems more reliable when quantifying samples with a low concentration of the target DNA.

    Similar quantification values across sperm samples suggest that all methods perform equally well to accurately and reliably measure concentration in relatively highly concentrated samples. This corresponds to the optimal scenario (Fig. 1, scenario A), with qPCR, dPCR and ddPCR all able to accurately and repeatedly quantify the target mtDNA across replicates. However, when the concentration of target DNA decreases, we observe variation in quantification values between the platforms and between replicates (Figs. 3 and 5). This is leading to a less optimal scenario (Fig. 1, scenario C), where variability increases and leads to inaccurate and non-repeatable quantification across replicates. As observed with the serial dilution, when the level of target DNA decreases even more, qPCR falls into the least optimal scenario, where target detection is stochastic across replicates and quantification is incorrect, leading to unreliable results (Fig. 1, scenario D). However, it should be noted that each of the three methods described in this study has its own strengths and weaknesses, making it important for researchers to choose the most appropriate approach based on specific goals, study questions and systems. Nonetheless, qPCR requires an additional step—a reference gene or a series of standard dilutions of

known concentrations, which are then used to estimate the concentration of target DNA. dPCR and ddPCR methods, on the other hand, eliminate the need for the standard curve, as the concentration values are readily available at the quantification endpoint based on Poisson statistics (*AMC Technical Brief, 2017*; *Quan, Sauzade & Brouzes, 2018*). As mentioned previously, both dPCR and ddPCR exhibited good amplification at low concentrations, and at the lowest concentration, ddPCR was more repeatable with lower CV of technical replicates. Nevertheless, it is important to consider that stochastic effects occur at low concentrations, potentially leading to the occurrence of false positive droplets. As in many laboratory processes, the human factor can be a potential limitation, and contamination or errors can happen. To mitigate this, it could be advised to employ an automated pipetting system, as it can reduce these effects, especially when working with low volume samples.

## CONCLUSIONS

In conclusion, the choice between the three different quantification platforms investigated in our study mainly relies on methodological tradeoffs which will vary depending on the research questions and sample types analyzed (Table 2). The qPCR platform has been used for decades and still is widely used across many fields of research due to its availability and broad applicability (*Bustin et al., 2009*). It is often more easily accessible than dPCR or ddPCR, as these more novel tools are more costly. When aiming to analyze samples with very high levels of target DNA, qPCR quantification will be the most appropriate technique to use and will show high repeatability and accuracy, while both dPCR and ddPCR platforms could be saturated by such highly abundant targets. However, when aiming to analyze moderately high levels of target DNA, we found that the use of any of the three investigated methods will lead to accurate and repeatable results, although ddPCR shows lower variation than both qPCR and dPCR. Finally, in the case of low target quantification, we recommend the use of either dPCR or ddPCR, as both platforms show reliable and similar quantification results, with ddPCR again showing significantly lower variation than dPCR.

## ACKNOWLEDGEMENTS

We would like to thank Birgitte Lisbeth Graae Thorbek, Audun Schrøder-Nielsen and Jarl Andreas Anmarkrud from the DNA lab at the Natural History Museum at the University of Oslo for their support and assistance in the lab. We also thank Marina Panova and SeAnalytics AB for processing the dPCR samples. We thank the three anonymous reviewers for their comments on a previous version of this article.

### Funding

Financial support was received from the Research Council of Norway (grant number 301592). The funders had no role in study design, data collection and analysis, decision to publish, or preparation of the manuscript.

### Grant Disclosures

The following grant information was disclosed by the authors:
Research Council of Norway: 301592.

### Competing Interests

The authors declare that they have no competing interests.

### Author Contributions

- Laima Bagdonaitė conceived and designed the experiments, performed the experiments, analyzed the data, prepared figures and/or tables, authored or reviewed drafts of the article, and approved the final draft.
- Erica H. Leder conceived and designed the experiments, performed the experiments, authored or reviewed drafts of the article, and approved the final draft.
- Jan T. Lifjeld conceived and designed the experiments, authored or reviewed drafts of the article, and approved the final draft.
- Arild Johnsen conceived and designed the experiments, authored or reviewed drafts of the article, and approved the final draft.
- Quentin Mauvisseau conceived and designed the experiments, performed the experiments, analyzed the data, prepared figures and/or tables, authored or reviewed drafts of the article, and approved the final draft.

### Animal Ethics

The following information was supplied relating to ethical approvals (*i.e.*, approving body and any reference numbers):

Sampling was conducted in adherence to ethical guidelines for use of animals in research and with permission from all relevant local authorities, and approved by the Norwegian Food Safety Authority (permit no. 23294 and 29575), and The Norwegian Environment Agency (permit no. 2021/39021).

### Data Availability

Information on samples used in the study (Tables S1–S3), measured LOD and LOQ values for synthetic DNA (Table S4), calculated copy number of synthetic mtDNA (Table S5) and Figures S1–S6 are provided in Supplementary information.docx.

Raw quantifiaction data is presented in 6 different sheets of Supplementary Table S6. xlsx.

Code used to analyze the data is provided in a word document named: RCode_methods_paper.docx.

### Supplemental Information

Supplemental information for this article can be found online at http://dx.doi.org/10.7717/ peerj.19278#supplemental-information.

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
