# Peer review of "Assessing reliability and accuracy of qPCR, dPCR and ddPCR for estimating mitochondrial DNA copy number in songbird blood and sperm cells"

_PeerJ, doi:10.7717/peerj.19278_

## Round 0.1 · original submission · Major Revisions

Please consider the detailed comments from the reviewers and submit a revised version along with a rebuttal letter.

Reviewer 1 ·

Basic reporting

The authors did a wonderful job with this work entitled “Assessing reliability and accuracy of qPCR, dPCR and ddPCR for estimating mtDNA copy number in songbird blood and sperm cells”. I have some suggestions for the betterment of this manuscript listed below:
1. Abstract: There are some ambiguities, please make it clear conceptual.
2. Keywords should be better than current words.
3. Latest citations missing (mainly in the introduction section), please add at least 20% of recent work.
4. Somewhere journal guidelines are missing, please rectify and fix it.
5. Some serious discussions about your novelty work are missing compared to others.

Experimental design

.

Validity of the findings

.

Reviewer 2 ·

Basic reporting

The manuscript is well written, the figures are prepared with the clear way so everything is easy to track.
The topic seems to be interesting for the international audience.

Experimental design

The experiment is well designed and visualized.
I have only some small comments:
Line 139. Provide the alignment method here, please.
Line 191. The samples are not mixed in droplet generator cartridges before the droplet generation. The cartridges have separate wells for sample and oil. Samples are mixed with oil during droplet generation process. Please rewrite this fragment properly.
Lines 246-257. It is possible to avoid oversaturation of droplets in case of preparing decimal dilutions of template material. The result have to be multiplied by dilution value. See paper: doi: 10.1016/j.psj.2024.104028.

Validity of the findings

This is a typical methodologic paper. However, quantification of genes or pathogenes genetic material in the samples with low amount of it is always risky and can be a reason of problems. This way the idea for this research is interesting and conclussions important. The paper points on strong point of ddPCR for evaluating a low amounts of gens of interest.

Additional comments

Some minor comments that have to be addressed before publication:
Line 69. change "calibration curves" into "standard curves", please.
Line 83. "...for target nucleic acids [refs].". You can add to this sentence:"...as well as screening for pathogen DNA and RNA in clinical samples with low amount of nucleic acids, like serum or swabs [refs:doi:10.1038/s41598-024-64587-3, doi:10.1371/journal.pone.0219175, doi: 10.1155/tbed/4684235 ].

The readers interested in using dPCR and ddPCR for evaluating the clinical samples could be also interested in your paper, so adding those few references and words could help with finding your paper and citing it in the future.

Reviewer 3 ·

Basic reporting

In this study the authors use various methods of PCR-based DNA quantification to evaluate the limit-of-detection (LOD) and limit-of-quantification (LOQ) for several different types of DNA samples (synthetic, avian blood, and avian sperm). The study seems to be a sound methodological assessment to identify the best PCR-based technique for mtDNA quantification; however, the motivation for why one would need to quantify mtDNA is not sufficiently addressed.

In general, the manuscript suffers from both grammatical and conceptual issues that make it a bit difficult to follow. The grammatical issues throughout can be easily addressed with some minor edits (see some example suggestions in the line-by-line comments). But overall, I believe the authors need to do some work on the overall structure of the narrative and background to improve the clarify the motivation for the study and make the story a bit a bit easier to follow.

Experimental design

One of the main confusions I have with the narrative is the use of synthetic and biologically derived DNA. I believe the idea was to test each method on a “control” of synthetic DNA for accuracy, then test precision on “real” biological samples. If so, I see the value in this approach, but it needs to be more clearly laid out in the manuscript. For example, why are the results of the synthetic DNA tests not included in the figures?

Moreover, there is some ambiguity whether the statistics are sound; this should be clarified or addressed. It is also my opinion that some of the figures need some work, and some thought should be given to what information they are trying to convey.

Validity of the findings

Findings seem reasonable. Once the overall clarity has been improved and the motivation sufficiently addressed, I believe this publication would likely be well suited for publication in this journal.

Additional comments

Line-by-line Comments:

22- Second sentence seems a little out-of-the-blue. Maybe not needed in the abstract? Or say before the bit where you introduce you used siskin-derived blood and sperm.

24- Style suggestion [SS]: Could drop the “While” and say “Previous studies..., but...” for clarity.

30- It’s a little confusing the way it’s presented here that you used synthetic then animal-derived DNA. Maybe clarify with a sentence before “Using synthetic DNA...” that says something like “We tested each method of these quantification methods on both synthetic and animal-derived DNA samples...”. Or add a transitional sentence before the siskin DNA bit to say “Then to confirm these results based on animal-derived DNA...”

33- SS: Could merge the last two sentences. “Our study provides... and indicates that...”

40- SS: Would move the “found in most...” clause to the end of the sentence.

52- SS: I personally don’t love “Interestingly” as a transition. Maybe there’s another one that would work here? Such as “Notably”?

54- “inter- and intraspecific” instead?

56- SS: “interest has been cast”, this phrasing is a little odd.

60- SS: Not sure “especially” is the right word here. “such as” would be better.

71- SS: “also known as”, should be “such as”.

83- Comma before “and”.

96- In general, the framing of the intro/methods story here is a little confusing. Why are you interested in the sperm characteristics and number of mitochondria in the siskin? A bit unclear to me at this point. I would suggest putting some more thought into the framing of the motivation of the study in the intro, then briefly introduce why the siskins were selected for the study species here in the methods. Specifically, for the intro, I suggest framing it thusly: (A) Why would anyone want/need to quantify the amount of mitochondrial DNA in a sample? (B) Why are bird sperm and blood specimens good test subjects for this? (C) Then get into the methods people use and the need to identify which is most appropriate, which is the driving motivation of this study. Next, when you introduce your study species in the methods it should feel less abrupt: abundant and easy to sample, background on the characteristics on the sperm / blood of this species, etc...

96- Also, given you first evaluate these methods using synthetic DNA (right?), it would be helpful to introduce that before methods about capturing and sampling the siskins. Specifically, I would move the “Primers and synthetic oligo design” up before capturing siskins to keep it chronological.

107- “micro-capillary” instead?

113 and 114- Should be no commas before these and’s here, right?

135- Isn’t the order Passeriformes?

128 to 282- I’m admittedly not an expert on qPCR-type methods in particular, but am generally familiar with them. The description of the PCR methods and results here appear to be sufficiently detailed.

232- Wickam et al. should be a numbered reference as the rest.

267- Report actual ANOVA p rather than the <0.05 throughout the results. Also, I’m not clear that ANOVA is being used correctly here. For each sample—sperm or blood (also, what about the synthetic DNA?)—you have three methods your comparing, qPCR, dPCR, and ddPCR. So, within the sample type you want to know if any pair of treatments vary significantly. Thus, you should run an ANOVA on each sample type to determine if any pair of the three methods has a significant difference, then follow up on those with ANOVA p < 0.05. I believe you should use a Tukey’s HSD test to determine which pairs of measurement techniques within the sample differ (rather than Student’s or Welsh’s t-tests). This should be important information for your discussion. If this is what you did, it’s not abundantly clear to me in the methods/results/discussion. So please clarify throughout.

283 to 339- Generally, discussion provides a clear and interesting assessment of the results.

340 to 355- Conclusions are reasonable.

Figure 1- I’m not sure this schematic adds any information to the study. Looks more like a slide you’d show in a presentation. Without the figure caption, I wouldn’t be able to glean anything from it. You could probably just drop it, but if you are going to include it, I think it would be more useful if it included: (1) descriptions of the mtDNA levels in the blood versus sperm in the image to justify why they’re both being used; (2) bullets summarizing the pros/cons of the different methods next to their image; (3) titles on each of the targets A-D as “precise and accurate”, “precise but inaccurate”, etc., to describe what they represent on the image.

Figure 2- “Data from seven dilution points (concentration range 2.50E-06 to 2.50E-12 (ng/μL)) were used to generate this figure.” Kind of odd phrasing. Maybe try: “The figure shows data from seven dilution points...” I may have missed something, but why did you select these seven dilution points for this figure? Also, why are the lines polynomial if you’d expect them to be linear as the dotted line? Do you address these points anywhere in the text?

Figure 3- Is a boxplot the correct way to represent this? Perhaps it is, but seems a little odd to lump all three technical replicates from 10 different samples and treat them as independent observations like this? Also, did you test these for significant difference using an ANOVA and then Tukey’s HSD? You should display asterisks/brackets to show if any varied significantly or state that they didn’t (see line comment for 267).

Figure 4- This is a handy figure. Might actually be better presented as a boxplot of CoV ~ Method to better show that qPCR has the lowest CoV. Also would be good to do ANOVA/Tukey on these as for the other comparisons.

Table 1- Why not just report the actual p-value? You could also have a column for significant Y/N.

---

## Round 0.2 · Major Revisions

Kindly examine the corrections noted by the reviewer, especially those related to synthetic DNA generation.

Reviewer 1 ·

Basic reporting

All suggestions and modifications are well applied.

Experimental design

All suggestions and modifications are well applied.

Validity of the findings

All suggestions and modifications are well applied.

Additional comments

All suggestions and modifications are well applied.

Reviewer 3 ·

Basic reporting

General Revision Comments:

The authors have done a nice job improving the narrative clarity on the motivation / need for the study and have addressed most critiques of the prior draft well.

One issue that remains outstanding for me: I am still a bit confused about the synthetic DNA. Specifically, it is not clear to me from the methods where or how this synthetic DNA was generated. I think this could be easily cleared up by adding a brief methods section after the bit about ‘passerine mtDNA primer design’ that explicitly explains the need for and creation of synthetic DNA in this study.

Once the following (mostly minor) comments are addressed, I think this study will be in good shape for publication in this journal:

Experimental design

Methods on the development of synthetic DNA seem to be absent.

Validity of the findings

N/A

Additional comments

Specific Revision Comments:

(Note: My line numbers correspond to the track-changes draft, not the PDF. My bad.)

Nice abstract! Flows well and lays out your study / key findings nicely.

Introduction is much clearer to me now and sets up your study well. Nice job.

189 to 214- Sorry, it’s possible I’m missing something, but where do you describe the “synthetic oligo design” here. Or do you just mean the primers by that? Later on, you mention synthetic mtDNA (line 321), but I can’t figure out where it came from. I think you should have a section on primer design (this section), then a separate section for the synthetic mtDNA you synthesized for testing—covering how and why you created and used the synthetic DNA before it comes up later in the qPCR section? [Circling back after finishing, this is my largest outstanding issue with the manuscript.]

190- A very brief topic sentence here would be helpful to start. Just concisely reiterating that you are designing a generic primer to amplify passerine mitochondrial DNA.

207- Pedantic, but I don’t think in vitro, in vivo, or in silico should have hyphens.

209- Because of the rearrangement, I’m not sure you’ve defined your “target species” yet? Maybe just include it in your list here and say “multiple songbird species” or something instead of “our target and other spp.”

210- Also, where did all these different samples come from? A museum tissue collection perhaps? Other studies from your lab? That info could be in Supp. Table 3, but a very brief summary would be useful here.

222- I concur, I think it’s good to include the trapping method here and that they were released, not collected.

252 to 346- Not commenting on the methods here as they’re out of my wheelhouse and appear reproducible.

321- See comment for ‘189 to 214’.

355- Again, I could be daft, but I don’t see anything in that section describing the synthetic DNA generation.

386- I’m a little confused here; if you used ANOVA/Tukey to test for significant differences, where do the Student and Welch’s t-tests come in? Please clarify which analyses were used for which comparisons.

390- You didn’t test on a non-passerine to confirm this did you? If not, you should say something like “this primer was found to amplify mtDNA across a range of passerine bird species” or whatever. Not clear that how you found it wouldn’t amplify mtDNA in a non-passerine if you didn’t test that.

398- Keep an eye on tense throughout. Should be “followed”.

400- Is the synthetic oligo the same as the synthetic DNA? Refer to it the same way throughout as to not introduce ambiguity.

406 to 416- Is this the section where student or Welch’s t-tests were used instead of ANOVA/Tukey, because you only have the two groups to compare? If so, please clarify that in the methods section. See comment for line 386.

407- “ranged”, “had slope from”, “had Y intercepts from”, “had R2 from”.

408- Pedantic, but all the DNA is “real”; say “synthetic and biologically-derived mtDNA”, or something to that effect.

498- Glad to see the p-values reported. Probably could truncate at three significant figures.

578- A table summarizing your conclusions/recommendations for when each method is/isn’t appropriate based on your findings could be very useful in your discussion/conclusions.

Fig 1- More useful now with the text describing differences in the sample types / your hypothetical outcomes. Should probably include your synthetic DNA in this schematic too with a description of what it was for, no?

Fig 2- Sorry to beat a dead horse, but again, please use synthetic oligo or synthetic DNA consistently throughout so that it’s clear they are not two separate things.

Fig 3- Two dependent clauses with “while” here. Change the second while to “and”.

Table 1- Would be nice to have a cell for which t-test (student/welch) was used in the table so it's easier tell. Could pick a consistent number of sig-figs to truncate p-values (see line comment 498).

---

## Round 0.3 · accepted · Accept

Thanks for addressing all the major issues. Your manuscript has now been accepted!

Reviewer 2 ·

Basic reporting

The manuscript was improved and now can be published.

Experimental design

No additional comments.

Validity of the findings

This is a typical methodologic paper. However, quantification of genes or pathogenes genetic material in the samples with low amount of it is always risky and can be a reason of problems. This way the idea for this research is interesting and conclussions important. The paper points on strong point of ddPCR for evaluating a low amounts of gens of interest.

Reviewer 3 ·

Basic reporting

Now that the methods on synthetic DNA generation are cleared up, I believe it's in good shape.

Experimental design

.

Validity of the findings

.

Additional comments

.